# Raising an Eye at Facial Muscle Morphology in Canids

**DOI:** 10.3390/biology13050290

**Published:** 2024-04-25

**Authors:** Courtney L. Sexton, Rui Diogo, Francys Subiaul, Brenda J. Bradley

**Affiliations:** 1Department of Population Health Sciences, Virginia-Maryland College of Veterinary Medicine, Virginia Polytechnic Institute and State University, Blacksburg, VA 24061, USA; 2Center for the Advanced Study of Human Paleobiology, Department of Anthropology, The George Washington University, Washington, DC 20052, USA; 3Department of Anatomy, Howard University School of Medicine, Washington, DC 20059, USA; 4Department of Speech, Language and Hearing Sciences, The George Washington University, Washington, DC 20052, USA

**Keywords:** canid morphology, facial muscles, domestication, facial expression, communication, dissection

## Abstract

**Simple Summary:**

Facial expressions are important for many animals when communicating with other individuals both within and outside of their own species or group. The canid family includes several species that are highly social. Dogs, in particular, have adapted, through domestication, facial expressions that aid in their social interactions with people. The presence of certain facial muscles facilitates these interactions, however it does not seem that these muscles are unique to dogs, but also present in other canids.

**Abstract:**

The evolution of facial muscles in dogs has been linked to human preferential selection of dogs whose faces appear to communicate information and emotion. Dogs who convey, especially with their eyes, a sense of perceived helplessness can elicit a caregiving response from humans. However, the facial muscles used to generate such expressions may not be uniquely present in all dogs, but rather specifically cultivated among various taxa and individuals. In a preliminary, qualitative gross anatomical evaluation of 10 canid specimens of various species, we find that the presence of two facial muscles previously implicated in human-directed canine communication, the *levator anguli occuli medialis* (LAOM) and the *retractor anguli occuli lateralis* (RAOL), was not unique to domesticated dogs (*Canis familiaris*). Our results suggest that these aspects of facial musculature do not necessarily reflect selection via human domestication and breeding. In addition to quantitatively evaluating more and other members of the Canidae family, future directions should include analyses of the impact of superficial facial features on canine communication and interspecies communication between dogs and humans.

## 1. Introduction

Anyone who has ever cohabitated with a dog knows “the look”—scrunched skin along a furry brow line accentuating two upward-curved half moons of eyes filled with puddles of iris. This so-called paedomorphic expression has been perfected by generations of dogs over millennia—some postulate it as an effective strategy to encourage human caregivers to do just that, give care [1]. But whether it is intentionally produced to solicit human empathy, or not, the ubiquity of “the look” among dogs is an evolutionary marvel.

Facial expression is an essential element in the human social repertoire [2,3,4,5] and dogs have adapted well to communicate with humans via both their own canid-specific behaviors, as well as those coopted from the humans with whom they cohabitate—including such facially-oriented signals as sustained eye contact/mutual gaze, raised eyes, head tilts, and others [6,7,8,9,10,11]. But are such cues resulting from and reserved for human-directed communication alone? “Body language” is known to be highly significant in canine communication among conspecifics and in interactions with non-kin, alike [12,13,14], and facial communication is also important for many other species [15,16,17,18,19,20,21,22,23,24,25,26].

M.W. Fox (1970) [27] describes the development of facial expressions in various canid species, including wolves, coyotes, grey foxes, red foxes, and Arctic foxes, finding facial expressions to be situation-specific (e.g., at play vs. aggressive), and associated with an increase or decrease in social distance between dyads. Notably, Fox also finds that wolves, coyotes, and domestic dogs differ from foxes in their wider range of simultaneous expressions, to which they attribute a possible “evolutionary advancement of visual signals in more social species” and that later-emerging components of facial expression complexity may be phylogenetically more recently acquired [27]. It has also been suggested that these differences in facial expressions mirror trends seen in primates, and align with the various canid species’ social organization and behaviors (e.g., cooperative breeding)—a significant exception being that domesticated dogs are not cooperative breeders, even those that are free-ranging.

Less is understood about how or if the nuanced and subtle facial expressions dogs recognize and respond to in humans translate for use or are relevant during communication among canine conspecifics.

Human facial expressions serve primarily to convey complex emotions, and the ability to produce them is directly related to evolved facial muscle morphologies [28,29,30]. For example, Powell et al. (2018) [31] describe trends in primate musculoskeletal evolution that suggest an increase in the number of head and neck muscles (necessary for enhanced complexity of expressions) precipitated a decrease in bone–muscle network density. They additionally find that the complexity of facial expressions in humans is related to modular integration/asymmetry of facial network modules, rather than more modular symmetry as in non-anthropoid primates.

While head/neck muscle diversity in primates is linked to functional differences (e.g., facial/vocal communication related to complex thoughts/emotions in humans), Powell et al. (2018) also stress that humans are not more complex than other primates in their overall morphology. As we are beginning to accept that dogs also experience rich emotional lives and can distinguish emotions in expressions [32,33,34], it is worth considering whether dogs, too, have undergone morphological changes related to functional adaptations of facial expressions, and, if so, how they compare to other canids [27].

Kaminski et al. (2017) [35] demonstrate that the quantity and variety of facial expressions produced by domestic dogs increases in response to attentive humans vs. non-attentive humans, with attention defined as the human experimenter facing toward the dog and making eye contact with no verbal or vocal signaling or cues. Using DogFACS, the Dog Facial Action Coding System [36], which identifies observable facial changes associated with underlying muscle movements measured by Action Units (AUs), those authors found that dogs display various movements of the eyes and areas of the face around the eyes with high frequency when in communication with people.

Concurrently, Burrows et al. (2018) [37] investigated whether the production of these varietal expressions in dogs is linked to changes in facial musculature over time, as determined by a comparison of facial muscles between wolves and domesticated dogs. According to those authors, “Gross results revealed that the dog and wolf samples were similar to one another in all facial musculature except for the levator anguli occuli medialis (LAOM) and the retractor anguli occuli lateralis (RAOL) muscles. All dog specimens routinely had these muscles while the wolf specimens varied in their presence and size” (Figure 1). Kaminski et al. (2019) [1] further suggested that this muscular anomaly, along with the propensity of dogs to produce a paedomorphic expression (e.g., “the look”) more often and with more intensity than wolves, suggests a muscular adaptation based on artificial selection via a human desire to nurture.

Questions remain as to whether this potential anatomical anomaly is a result of domestication and selection, shaped by centuries-worth of human-directed communication on the part of *Canis familiaris*. Given that intentional breeding has produced a broad range of relative eye muscle sizes in various domesticated dog breeds, it is likely that other factors are at play, and facial communication is complex. Neuronal signaling to the muscles in question, and surface-level physical facial pigmentation related to domestication and neural crest cell migration also impact the execution and effectiveness of facial gesturing [38,39].

But understanding the degree to which sociality and communication in the context of domestication can be considered drivers of anatomical change in canids has critical implications across research fields. Such information could be useful in corroborating conflicting DNA evidence in attempting to define speciation events [40,41,42,43]. It could also be illuminating in identifying how social pressures contribute to ecological niche exploitation and population movement patterns, such as for coyotes inhabiting dense urban areas [44], as well as long-term population viability for threatened and endangered species among whom introgression is commonly a necessity, such as the highly endangered red wolves in the Southeastern US [45]. From an anthropological viewpoint, canid faces can even potentially offer clues to the emergence of certain traits in early humans [46]; but see [47].

Here, we explore whether adaptive communication strategies, which emerged early in dogs concurrently with or as a byproduct of the domestication process [8,48], may have impacted the evolution of morphologically distinct facial muscles in dogs, especially those around the eye. Specifically, the aim of this opportunistic study was to employ gross, qualitative analyses of facial anatomy in several *Canidae* species, with a focus on the two muscles in the eye region (LAOM and RAOL) that have been previously implicated in facilitating interspecies communication between humans and domesticated dogs. Our goal was to make a preliminary determination based on limited data from an opportunistic sample of whether the LAOM and RAOL are anatomically unique to domesticated dogs or are also present in other canid species. Due to the relatively recent divergence of the species in question, we hypothesize that these muscles will be present across species.

**Figure 1 biology-13-00290-f001:**
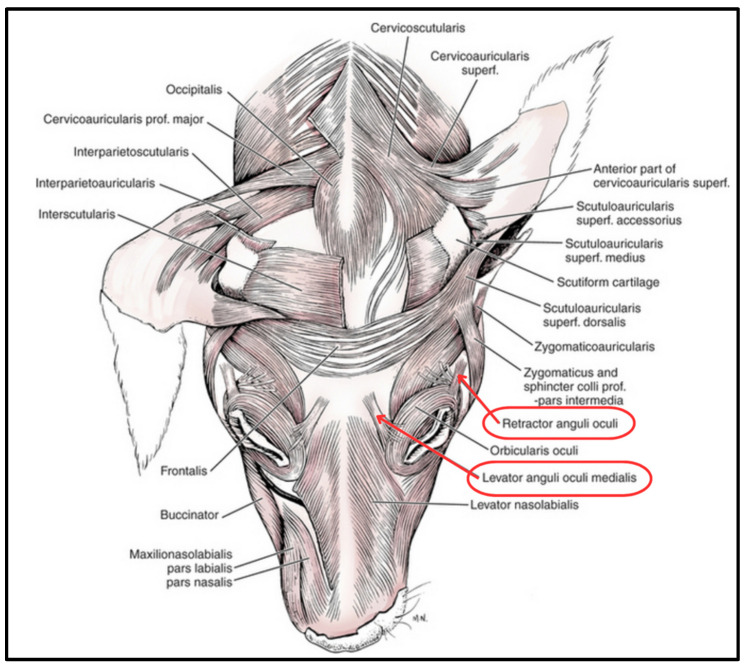
Facial muscle anatomy of the domesticated dog. Two muscles in the eye region, the RAOL and the LAOM, are implicated in facial communication. (Modified from [49]).

## 2. Materials and Methods

### 2.1. Ethical Considerations

Anatomical evaluation was conducted under the supervision of Dr. Rui Diogo, Associate Professor of Anatomy at the Howard University School of Medicine. Canid specimens were collected opportunistically from various organizations and individuals, including local and state animal control facilities and state fish and game agencies between October 2017 and February 2019. No live animal subjects were included in this, nor were any of the animals’ lives taken for the express purpose of this research. This study is therefore exempt from IACUC approval.

### 2.2. Anatomical Specimens and Experimental Procedure

No live animal subjects were included in this study, and none of the subjects’ lives were taken for the express purpose of this research. The domesticated dogs (*N* = 2) had been recently euthanized at the Frederick County Animal Control facility and were donated by Dr. Virginia Pierce. Coyote (*N* = 3) and fox (*N* = 5) specimens (see below) were collected from state-coordinated culling and take events and permitted hunters, with the assistance of USDA (APHIS) wildlife biologist, Kyle Van Why (PA).

Dissection procedure: All specimens were stored at −20 °C in the Rui Diogo lab for comparative anatomy at the Howard University College of Medicine. Under the supervision of Dr. Rui Diogo, standard dissection procedures were followed [50] to evaluate the facial muscles of the predeceased canids including: two domestic dogs (*Canis familiaris*); three coyotes (*Canis latrans*), two from the Eastern US and one from the Pacific Northwest US; two gray foxes (*Urocyon cinereoargenteus*); two red foxes (*Vulpes vulpes*); and one Arctic fox (*Vulpes lagopus*).

Laboratory surfaces were pre-cleaned with 10% bleach solution. Bench paper was placed under dissecting trays, and specimens were placed on dissection trays atop laboratory tissue paper. Scalpel sharpness was tested, and knives and additional tools were cleaned with 10% bleach solution prior to each use. Investigators wore gloves, lab coats, glasses and masks when handling specimens. Cutting, pinning, and tagging followed methods described by Diogo et al. (2008) [50]. Specimens were fresh or frozen fresh when dissected. All surfaces and tools were cleaned post-use, and remaining specimens were freezer-stored in biohazard bags.

Specimens were analyzed for the presence, absence, and relative robustness/gracility of two target ocular region facial muscles, the levator anguli oculi medialis (LAOM) and the retractor anguli oculi lateralis (RAOM) (Figure 1). Photographs were taken for additional, post-laboratory analysis and comparison between specimens (see Appendix A).

Because no wolf (*Canis lupus*) specimens were obtainable during the period of the study, data from Burrows et al. (2018) [37] serve as a reference for that species.

## 3. Results

Gross qualitative examination revealed differences in the presence, absence, and relative robustness of the LAOM and the RAOL in accordance with the following observations and as summarized in Table 1.

Specimen 1: Dog (adult, male), from the Frederick County Animal Control facility in Maryland, was a Chihuahua breed, weighing approximately 5lbs. Both target muscles were present and robust (Appendix A).

Specimen 2: Dog (adult, male), from the Frederick County Animal Control facility in Maryland, was a mixed breed, identified as part Pitbull Terrier, weighing approximately 55lbs. Both target muscles were present and robust (Appendix A).

Specimen 3: Coyote (young adult, female), collected from Lake County, Oregon, a rural but human-inhabited area. Both target muscles were present. RAOM appeared slightly less robust than in the dogs, though the general facial structure was overall less fleshy, and a matter of relative tissue density could account for the difference (Appendix A).

Specimen 4: Coyote (adult, female), collected from rural southeastern Pennsylvania. RAOM is obviously present and robust on the left side. The left LAOM is difficult to distinguish, though there does seem to be some muscle mass present. The right side of head was severely degraded due to damage during defrosting after shipment to the lab (Appendix A).

Specimen 5: Coyote (adult, male), collected near the Pittsburgh, Pennsylvania airport, a human-populated area. The LAOM is present; RAOM also present, perhaps slightly less robustly than in the dogs (Appendix A).

Specimen 6: The Arctic fox specimen was previously dissected by Dr. Diogo and slightly degraded. Age and sex are unknown. It is possible that the RAOM may be present, but difficult to determine. A muscle similar in robustness to the LAOM appears present more ventrally and is connected to the orbicularis oculi (Appendix A).

Specimen 7: Red fox (adult, female), collected from Delaware County, Pennsylvania, a human-populated area. Specimen does not appear to have either target muscle. There is a possibility that the RAOM is present, but this muscle structure appears as more an extension of the *orbicularis oculi*, wrapping dorsally toward the ear, as opposed to a distinct muscle as presented in the domesticated dog specimens (Appendix A).

Specimen 8: Red fox (juvenile, male), collected from Philadelphia County, Pennsylvania, an urban area. Specimen does not appear to have either target muscle. As in the adult female, there is a possibility that the retractor is present, but again it looks like it is more an extension of the orbicularis oculi. The orbicularis oculi also extends distal laterally where the RAOM is present in the dogs (Appendix A).

Specimen 9: Gray fox (adult, female), collected from Harrisburg, PA, a human-populated area. This specimen had very tough skin and very little tissue between the skull and the skin compared to the other species. The LAOM does not seem to be distinct from the orbicularis oculi; if it is, it is very weakly presented. No RAOM appears distinct from the orbicularis oculi. This specimen presents interesting asymmetries in the facial tissues: the left side of the face appears to be damaged—possibly post-mortem—the result actually being a clearer presentation of the muscles as some of the other tissues are flattened; the LAOM does not appear to be present, especially on the right (Appendix A).

Specimen 10: Gray fox (adult, male), collected from Harrisburg, PA, a human-populated area. As with the female specimen of the same species, this specimen had very tough skin and very little tissue between the skull and the skin compared to the other species. The LAOM does not seem to be distinct from the orbicularis oculi; if it is, it is very weakly presented. No RAOM appears distinct from the orbicularis oculi (Appendix A).

No apparent sex differences were observed.

No differences in relative robustness between the dog breeds were observed.

Of note: Thin, striated micro-muscles extending around the brow ridge and below the eye socket, connecting the LAOM to the RAOM were observed in both dog specimens, but not observed in other species (Appendix A).

## 4. Discussion

Given the relatively recent divergence (~12 my) from a last common ancestor (LCA) for the least-related of the species analyzed [51] and the potential range of functions of the muscles evaluated [52], it is unremarkable that the present study finds the levator anguli oculi medialis (LAOM) and the retractor anguli oculi lateralis (RAOL) to be present in varying degrees of robustness across species, as we hypothesized. The ability to adjust the skin of the face around the eyes could aid in visual acuity, and the muscles could be associated not exclusively with a social context, but with eye movement in general, as described by [53].

According to Burrows et al. (2018) [37], both the LAOM and the RAOL were found present in wolf specimens, though without consistency and with varied robustness. Those authors also note that at least one dog lacked one of the muscles. From the present study, three of the most notable results include: (1) the absence of both muscles in wild foxes; (2) the presence, fairly consistently, if slightly more gracile, of both muscles among the coyotes; and (3) the presence of the smaller, striated muscles observed along the brow ridge in the dog specimens.

One of the most significant challenges to this study is the high degree of phenotypic diversity in domestic dogs as a result of human-directed breeding [54,55,56]. It is to be expected that the musculature of a toy poodle, for example, would be grossly different from that of a mastiff. At its worst, breeding for certain morphologies such as brachycephaly and dolichocephaly in domesticated dogs can convey functional disadvantages, including alteration and/or elimination of muscles implicated in mimicry [57].

More distant relatedness could possibly account for the difference between the foxes and the other species observed, as could lack of interaction with humans (compared to coyotes and dogs), the latter of which would support the Kaminski et al. (2019) [1] hypothesis that the muscles’ presence in dogs is an adaptation for communication with humans. If interaction is the mechanism at play, it would make sense to see the muscles present in the wolf specimens, as the individuals included in the Burrows et al. (2018) [37] study were raised in captivity. The difficulty in determining the presence/absence in the foxes (distinguishing distinct muscles from extensions of the orbicularis was particularly challenging with the foxes) could also expand on that hypothesis by suggesting that these muscles are basal features that are “flexed” with use, or diminished/degraded with nonuse.

The consistent presence of both target muscles with less robustness in the coyotes likewise offers support to this hypothesis. As social learners, coyotes in the Americas potentially represent an interesting case of self-domestication, or at the very least habituation, in progress. In the United States, coyotes’ range extends across the majority of the country and they can be found in both rural and urban areas alike, with an increasing presence being noted in urban environments [58]. Coyotes are more habituated than wolves, with their occupied habitat significantly overlapping human-occupied areas [59,60,61].

Recent studies show that coyotes inhabiting urban environments are bolder and more exploratory (though an important distinction to make here is that bold does not necessarily equate with aggressive, as is sometimes misconstrued) [62], and that offspring of more habituated parents also show less fear of humans [63]. There is also significant admixture between coyotes and domestic dogs in North America (especially in the eastern U.S. and Canada), as well as coyotes and wolves, which could impact muscular presentation if indeed the association with domesticated dogs is valid [44,64,65,66].

Additional behavioral data, specifically DogFACS-related, will be necessary to determine whether the presence of the muscles in this species is specifically related to selective pressures from humans based on facial expressions/communication, or, alternatively, more generally a part of the suite of physical changes accompanying domestication/domestication-adjacent behavioral adaptations. Even if the latter is the case, however, the robustness of the muscles in the dogs could be a result of adapted innervation patterns and new and regular use of existing structures.

Another alternative hypothesis is that the muscles are related to sociality and within-group cooperation, generally. Wolves, coyotes, and dogs are the most social of the species evaluated, and all presented with both muscles, while gray foxes are more solitary and did not present with either [67,68,69]. Of course, this study was conducted with a limited sample size, a comparison of a narrow range of specimens, and is qualitative in design. Future studies should quantify muscle robustness and also control for body size in analyses. Continued research should likewise include anatomical analysis of wild versus captive wolves, and especially captive, tamed foxes, as we have seen significant behavioral adaptations in the domesticated foxes [70,71,72].

To get a more complete picture across the family, additional specimens should also include free-ranging/street dogs, dingoes, and African wild dogs. Like the captive foxes serving as a foil to the wild foxes examined herein, free-ranging dogs would also be of particular interest, because while genetically the same as tame “pet” dogs, they experience very different social interactions with humans, and with each other [73,74].

There are also additional issues of behavior to consider. The complex hierarchical social structures critical for survival in wolves [75,76,77] also eliminate any demand for wolves to interact with humans that is present in household-living and even free-ranging domesticated dogs. Wolves who have been the subjects of behavioral studies such as those measuring performance on various cognitive competency and cooperation tasks, however, have all been raised in captivity, which inherently alters their response to humans (which may be helpful in terms of supporting the theory of communication as a driver for morphological change). But because wolves in the wild do not communicate with humans, it would be pertinent to evaluate what meaningful expressions may be used among wolves between conspecifics, which may require the same eye-region muscles implicated here.

## 5. Conclusions

Exploring the significance of learned and/or acquired behaviors on the function of physical and anatomical attributes from species to species is a key evolutionary inquiry. In particular, identifying the same muscles present across taxa being used (or not used) for different purposes among species and even individuals has the potential to inform our understanding of neurological development and cognitive function—i.e., does presence equate to potential?

Because the present study’s results offer preliminary indications that the anatomical attributes in question (the two targeted facial muscles LAOM and RAOL) are present—albeit with variation—throughout the canid family, additional investigation should proceed in two directions:Analyses of neural differences as they relate to behavioral differences acquired during domestication. In particular, are dogs’ brains now “wired” differently from wolves or other canids to recognize facial expressions in humans (and perhaps less so in conspecifics), and to instinctively make similar expressions to communicate with humans on whom they are dependent [48,78]?Analyses of the impact of superficial phenotypic features of the face—such as pigmentation, markings, and patterning—on interspecies communication, specifically between dogs and humans. Previous work suggests that the diversity of facial appearance (e.g., superficial facial features) is significant in conspecific communication in highly social species of primates and canids [11,79,80], and perhaps even more meaningful than subsurface muscle movements.

## Figures and Tables

**Table 1 biology-13-00290-t001:** Presence, absence, and relative robustness of target muscles around the eye in specimens analyzed. Y = present; N = not present; n (orbicularis?) = some robustness in the region but does not appear to be separate from orbicularis oculi.

Specimen	Species	~Age/Sex	LAOM Present? Y/N	RAOL Present? Y/N	Social/Reproductive Behavior
N/A	Gray wolf (*C. lupus*)	---	Burrows et al., 2018 [37] describe wolf sample (*N* = 4) in which presence and size of both muscles varied.		Social/cooperative breeders
1	Dog (*C. familiaris*)	Adult/M	Y (robust)	Y (robust)	Social/no cooperative breeding
2	Dog (*C. familiaris*)	Adult/M	Y (robust)	Y (robust)	Social/no cooperative breeding
3	Coyote (*C. latrans*)	Young Adult?/F	Y (gracile)	Y (gracile)	Social/communal—den sharing
4	Coyote (*C. latrans*)	Adult/F	Y (gracile)	Y	Social/communal—den sharing
5	Coyote (*C. latrans*)	Adult/M	Y (robust)	Y (gracile)	Social/communal—den sharing
6	Arctic fox (*V. lagopus*)	unk/unk	N	undetermined	Monogamous pairs/some den sharing
7	Red fox (*V. v. fulvus*)	Adult/F	N	n (orbicularis?)	Social/polygynandry
8	Red fox (*V. v. fulvus*)	Juvenile/M	N	n (orbicularis?)	Social/polygynandry
9	Gray fox (*U. cinereoargenteus*)	Adult/F	N	N	Solitary/seasonal monogamy/ not cooperative
10	Gray fox (*U. cinereoargenteus*)	Adult/M	N	N	Solitary/seasonal monogamy/ not cooperative

Note: Specimens highlighted in green suggest high certainty of muscle presence. Specimens highlighted in yellow suggest a lower level of certainty in regard to presence of one or both of the muscles—this is indicated by a lowercase Y/N (y/n).

## Data Availability

Available data are included in the text and Appendix A.

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
