# Peer review of "Raising an Eye at Facial Muscle Morphology in Canids"

_biology, 2024, doi:10.3390/biology13050290_

Round 1

Reviewer 1 Report

Comments and Suggestions for Authors

The manuscript addresses a relevant topic and is generally well-written with appropriate methodological construction. However, there are some parts where improvements are necessary.

-The introduction is overly lengthy and occasionally delves into aspects not directly related to the central theme of the article. I recommend condensing it and ensuring that it clearly highlights the innovative aspects of the study. It would be beneficial to explicitly discuss whether similar studies exist in the literature and emphasize the unique contributions of the research if it is entirely novel.

-Ethical considerations warrant attention. The manuscript does not mention whether the study underwent evaluation and approval by an ethics committee. If such an evaluation took place (or not), it is imperative to clearly state and justify this within the manuscript.

-Considering the importance of comparative analyses, I suggest evaluating the inclusion of some of the supplementary images within the main body of the article.

-Methods of dissection should either be presented or appropriately referenced.

-Regarding the abbreviation "LCA," it should be defined within the text or avoided if possible to ensure clarity for readers.

-Considering the discussion, it would be beneficial to further elaborate on the implications of the study’s findings for understanding the evolutionary and behavioral aspects of the studied species.

Author Response

The manuscript addresses a relevant topic and is generally well-written with appropriate methodological construction. However, there are some parts where improvements are necessary.

  • Thank you for taking the time to review our manuscript and provide feedback. 

The introduction is overly lengthy and occasionally delves into aspects not directly related to the central theme of the article. I recommend condensing it and ensuring that it clearly highlights the innovative aspects of the study. It would be beneficial to explicitly discuss whether similar studies exist in the literature and emphasize the unique contributions of the research if it is entirely novel.

  • Thank you for this recommendation. As this piece is relevant to a larger body of work regarding domestication and canine behavior, we think it is important to include background that may not directly relate to the dissection and wish to keep it included. We have discussed previous work that inspired this study (Lines 92-110).

Ethical considerations warrant attention. The manuscript does not mention whether the study underwent evaluation and approval by an ethics committee. If such an evaluation took place (or not), it is imperative to clearly state and justify this within the manuscript.

  • We have updated and made explicit our ethical considerations statement to the manuscript (Lines 156-158)

Considering the importance of comparative analyses, I suggest evaluating the inclusion of some of the supplementary images within the main body of the article.

  • We do not think that the quality of the photographs (which were taken to be used as reference, not necessarily for publication) is not ideal for main text figures. The images were taken to support our notes, and we therefore opted to include them in supplemental materials as a reference point for readers.

Methods of dissection should either be presented or appropriately referenced.

  • We have clarified and referenced dissection procedures (Lines 168-181).

Regarding the abbreviation "LCA," it should be defined within the text or avoided if possible to ensure clarity for readers.

  • We have defined this acronym in the text. 

Considering the discussion, it would be beneficial to further elaborate on the implications of the study’s findings for understanding the evolutionary and behavioral aspects of the studied species.

  • We have included discussion of these implications (Lines 307-313; 323-348).

Reviewer 2 Report

Comments and Suggestions for Authors

Very valuable results that indicate the specificity of facial expressions and control muscles in canidae species, which are social to varying degrees. An interesting discovery regarding the micromuscles that connect both analyzed muscles. A valuable element of the work is also the indication of directions for further research that can be carried out in the context of the influence of relationships with humans on the structure and functioning of facial muscles in canids. It is worth continuing research in the directions described in the manuscript.

Author Response

Very valuable results that indicate the specificity of facial expressions and control muscles in canidae species, which are social to varying degrees. An interesting discovery regarding the micromuscles that connect both analyzed muscles. A valuable element of the work is also the indication of directions for further research that can be carried out in the context of the influence of relationships with humans on the structure and functioning of facial muscles in canids. It is worth continuing research in the directions described in the manuscript.

  • Thank you very much for taking the time to review our manuscript and provide feedback. We appreciate your encouragement and are looking forward to seeing this paper published.

Reviewer 3 Report

Comments and Suggestions for Authors

This study provides an analysis of anatomical variations in facial muscles within the canid family and their potential impact on communication and domestication. Including three canid species (coyotes, dogs, foxes) offers a good basis for understanding facial muscle adaptations. This approach allows for exploring the evolution of physical characteristics in relation to domestication and interspecies communication.

However, the study could benefit from deeper analysis on whether the results are representative of the general population of each studied species, especially given the focus on a limited range of specimens. This is critical for the robustness of the overall conclusions. Expanding the sampling to include a wider variety of individuals within each species would be beneficial. 

More importantly, the quality of the samples, the dissections, and the pictures shown is far from ideal, which further complicates the interpretation of the data and the robustness of the findings, and generates doubts about the validity of the conclusions drawn.

The study's reliance on images of exceptional cases with poor quality significantly undermines the reliability of observational conclusions.

Author Response

This study provides an analysis of anatomical variations in facial muscles within the canid family and their potential impact on communication and domestication. Including three canid species (coyotes, dogs, foxes) offers a good basis for understanding facial muscle adaptations. This approach allows for exploring the evolution of physical characteristics in relation to domestication and interspecies communication.

However, the study could benefit from deeper analysis on whether the results are representative of the general population of each studied species, especially given the focus on a limited range of specimens. This is critical for the robustness of the overall conclusions. Expanding the sampling to include a wider variety of individuals within each species would be beneficial. 

More importantly, the quality of the samples, the dissections, and the pictures shown is far from ideal, which further complicates the interpretation of the data and the robustness of the findings, and generates doubts about the validity of the conclusions drawn.

The study's reliance on images of exceptional cases with poor quality significantly undermines the reliability of observational conclusions.

  • Thank you for taking the time to review our manuscript and provide feedback. While we recognize that expanding the sample size and variation of species included would contribute to more robust results, the specimens analyzed in this study were collected opportunistically and are meant to serve as a preliminary sample, only (Line 138). We agree that follow-up studies should include additional individuals. This preliminary, qualitative data description is not intended to be comprehensive, rather to open the door to further investigation of potentially important elements of canid evolution relative to communication and social dynamics (323-334). We likewise recognize that the quality of photos is not ideal for main text figures. The images were taken to support our notes, and are included in supplemental materials as a reference point for readers.

Reviewer 4 Report

Comments and Suggestions for Authors

Dear Authors,

I really appreciate yours tremendous effort you did in this study to submit the manuscript. By my standpoint, you worked hard but the manuscript to be accepted for publication I consider reasonable to be re-write properly the hypothesis   of this study.

Author Response

Dear Authors,

I really appreciate your tremendous effort you did in this study to submit the manuscript. By my standpoint, you worked hard but the manuscript to be accepted for publication I consider reasonable to be re-write properly the hypothesis of this study, and if yours upfront expectations from the study purpose are confirmed or not by your results.

The hypothesis is a scientific and clever guess, which should be investigated, based on collected data, and then confirmed or rejected.

  • Thank you very much for taking the time to review our manuscript and provide feedback. We have explicitly stated our hypothesis in the introduction (Line 140), and have also referenced it in the discussion (Line 270).

Round 2

Reviewer 3 Report

Comments and Suggestions for Authors

While the preliminary nature of the study and the opportunistic collection of samples are understandable, the results presented do not reach the level of quality and rigor expected. I would encourage, as you yourself suggest, the conduct of additional studies which could provide the foundation for a future publication.

I have no doubt about the interest and relevance of your study. In fact, a significant contribution has recently appeared in this same field of research: Smith HF, Felix MA, Rocco FA, Lynch LM, Valdez D. 'Adaptations to sociality in the mimetic and auricular musculature of the African wild dog (Lycaon pictus),' published in Anat Rec (Hoboken) on April 10, 2024.